# DISTRIBUTED ZEROTH-ORDER OPTIMIZATION: CONVERGENCE RATES THAT MATCH CENTRALIZED COUNTERPART

## ABSTRACT

Zeroth-order optimization has become increasingly important in complex optimization and machine learning when cost functions are impossible to be described in closed analytical forms. The key idea of zeroth-order optimization lies in the ability for a learner to build gradient estimates by queries sent to the cost function, and then traditional gradient descent algorithms can be executed replacing gradients by the estimates. For optimization over large-scale multi-agent systems with decentralized data and costs, zeroth-order optimization can continue to be utilized to develop scalable and distributed algorithms. In this paper, we aim at understanding the trend in performance transitioning from centralized to distributed zeroth-order algorithms in terms of convergence rates, and focus on multi-agent systems with time-varying communication networks. We establish a series of convergence rates for distributed zeroth-order subgradient algorithms under both one-point and two-point zeroth-order oracles. Apart from the additional node-to-node communication cost due to the distributed nature of algorithms, the established rates in convergence are shown to match their centralized counterpart. We also propose a multi-stage distributed zeroth-order algorithm that better utilizes the learning rates, reduces the computational complexity, and attains even faster convergence rates for compact decision set.

## 1 INTRODUCTION

Various machine learning tasks ultimately boil down to solving optimization problems of different forms, where the cost functions are formed jointly by the data accumulated in experiences and the model used in representing the learning framework. Gradient descent algorithms have been playing a foundational role in practically solving such optimization problems. However, for learning tasks with high-dimensional data and involved learning representations, access to the gradient of the cost function may turn out not possible: the cost function supporting the learning may not have a closed analytical form; or it is simply too computationally costly to be properly differentiated. Zeroth-order optimization provides a systemic way of facilitating gradient descent without direct access to gradient information, where oracles query the cost function values and generate gradient estimates. Zeroth-order methods have shown a number of successful applications, e.g., searching for adversarial attacks in deep learning Chen et al. (2019); Liu et al. (2019) and policy search in reinforcement learning Vemula et al. (2019).

The literature has also explored the potential in extending the standard (centralized) zeroth-order optimization to distributed settings over multi-agent systems, where the data and cost functions are scattered across a network of decentralized agents. With the help of a communication network, the agents may collaboratively solve the network-level optimization task by iteratively exchanging decisions obtained from local zeroth-order descent. The rates of convergence of centralized zeroth-order optimization algorithms are now well understood for several sub-classes of convex functions. We are interested in systematically investigating these convergence rates scale for the corresponding distributed algorithms, and focus on the case of time-varying communication networks.

## 1.1 PROBLEM DEFINITION

Consider a network of agents (nodes) $\mathcal{V} = \{1, \ldots, N\}$. The agents aim to collectively solve the following distributed optimization problem

$$
\begin{aligned}
\text{minimize} \quad & f(\mathbf{x}) := \sum_{i=1}^{N} f_i(\mathbf{x}) \\
\text{subject to} \quad & \mathbf{x} \in \mathcal{X}.
\end{aligned}
\tag{1}
$$

Here $\mathbf{x} \in \mathbb{R}^d$ is the decision variable, $\mathcal{X} \subseteq \mathbb{R}^d$ is a convex decision space, and $f_i : \mathbb{R}^d \to \mathbb{R}$ is a private convex objective function associated with agent $i$.

The communication network connecting the nodes is described by a time-varying graph $\mathcal{G}(t) = (\mathcal{V}, \mathcal{E}(t))$, where $\mathcal{E}(t)$ is the set of activated links at time $t$. Let $\mathbf{A}(t)$ be a weight matrix at time $t$ for the graph $\mathcal{G}(t)$: for each link $(i, j) \in \mathcal{E}(t)$, a weight $[\mathbf{A}(t)]_{ij} > 0$ is assigned, and $[\mathbf{A}(t)]_{ij} = 0$ for $(i, j) \notin \mathcal{E}(t)$. We impose the following assumption on the communication network $\mathcal{E}(t)$ and the weight matrix $\mathbf{A}(t)$.

**Assumption 1** *(i) There exists a positive integer $B$ such that the union graph $(\mathcal{V}, \mathcal{E}(kB+1) \cup \cdots \cup \mathcal{E}((k+1)B))$ is strongly connected for all $k \geq 0$; (ii) $\mathbf{A}(t)$ is doubly stochastic, i.e., $\sum_{i=1}^{N}[\mathbf{A}(t)]_{ij} = 1$ and $\sum_{j=1}^{N}[\mathbf{A}(t)]_{ij} = 1$; (iii) $[\mathbf{A}(t)]_{ii} \geq \xi$ for all $i$, and $[\mathbf{A}(t)]_{ij} \geq \xi$ if $(j, i) \in \mathcal{E}(t)$, where $\xi > 0$.*

## 1.2 FUNCTION CLASSES

Let $\mathcal{F}_{\mathsf{cvx}}$ denote the set of all convex functions on $\mathbb{R}^d$. We define the following three classes of convex functions in $\mathcal{F}_{\mathsf{cvx}}$.

- The Lipschitz continuous class $\mathcal{F}_{\mathsf{lip}}(L_f, \mathcal{X})$ contains the functions in $\mathcal{F}_{\mathsf{cvx}}$ that admit a finite Lipschitz constant $L_f$ over $\mathcal{X}$, i.e.,

$$
\mathcal{F}_{\mathsf{lip}}(L_f, \mathcal{X}) := \{g \in \mathcal{F}_{\mathsf{cvx}} : \ \forall \mathbf{x}, \mathbf{x}' \in \mathcal{X}, |g(\mathbf{x}) - g(\mathbf{x}')| \leq L_f \|\mathbf{x} - \mathbf{x}'\|\}.
$$

- The smooth class $\mathcal{F}_{\mathsf{smo}}(s_f, \mathcal{X})$ contains the functions that admit a $s_f$-Lipschitz continuous gradient over $\mathcal{X}$, i.e.,

$$
\mathcal{F}_{\mathsf{smo}}(s_f, \mathcal{X}) = \{g \in \mathcal{F}_{\mathsf{cvx}} : \ \forall \mathbf{x}, \mathbf{x}' \in \mathcal{X}, \|\nabla g(\mathbf{x}) - \nabla g(\mathbf{x}')\| \leq s_f \|\mathbf{x} - \mathbf{x}'\|\}.
$$

- The strongly convex class $\mathcal{F}_{\mathsf{sc}}(\mu_f, \mathcal{X})$ contains the functions that are $\mu_f$-strongly convex, i.e.,

$$
\mathcal{F}_{\mathsf{sc}}(\mu_f, \mathcal{X}) = \left\{g \in \mathcal{F}_{\mathsf{cvx}} : \forall \mathbf{x}, \mathbf{x}' \in \mathcal{X}, g(\mathbf{x}) \geq g(\mathbf{x}') + \langle \nabla g(\mathbf{x}'), \mathbf{x} - \mathbf{x}' \rangle + \frac{\mu_f}{2}\|\mathbf{x} - \mathbf{x}'\|^2\right\}.
$$

## 1.3 CONTRIBUTIONS AND RELATED WORK

**Contributions**. We first present MAZOPA, a multi-agent zeroth-order projection averaging algorithm. In MAZOPA, the agents iteratively carry out local zeroth-order descents for their private costs to generate intermediate decisions, send these intermediate decisions to their neighbors over the graph $\mathcal{G}(t)$, and then update their decisions by projecting the average neighboring intermediate decisions onto $\mathcal{X}$. For distributed zeroth-order oracles based on one-point or two-point estimates, a series of convergence rate results are established for the three basic function classes. Remarkably, the convergence rates for distributed algorithms are found to be matching their centralized counterpart, and sometimes even tighter rates are obtained, as summarized in Table 1. These results show that by paying the price of node-to-node communication, distributed zeroth-order optimization provides equal performance guarantees as those of centralized approaches. Next, we generalize the MAZOPA to a multi-stage setting, where the local zeroth-order descents take place for multiple steps before the projected averaging in a sequence of epochs. Such multi-stage MAZOPA is shown to be able to reduce the computational complexity, while providing improved convergences rates compared to MAZOPA when the decision set is compact.

Table 1: Convergence rates established for MAZOPA

| ZOO | Lipschitz Class [Centralized Counterpart] | Strongly Convex Class [Centralized Counterpart] |
|---|---|---|
| One-point | $\mathcal{F}_{\mathsf{lip}}$: $\mathcal{O}\big(\frac{d}{T^{1/4}}\big)$ $\big[\mathcal{O}\big(\frac{d}{T^{1/4}}\big)$ Flaxman et al. (2005)$\big]$ | $\mathcal{F}_{\mathsf{lip}} \cap \mathcal{F}_{\mathsf{sc}}$: $\mathcal{O}\big(\frac{d^2}{T^{1/3}}\big)$ $\big[$ $\mathcal{O}\big(\frac{(d^2 \ln(T))^{2/3}}{T^{1/3}}\big)$ Agarwal et al. (2010) $\big]$ |
| | $\mathcal{F}_{\mathsf{lip}} \cap \mathcal{F}_{\mathsf{smo}}$: $\mathcal{O}\big(\frac{d}{T^{1/3}}\big)$ | $\mathcal{F}_{\mathsf{lip}} \cap \mathcal{F}_{\mathsf{smo}} \cap \mathcal{F}_{\mathsf{sc}}$: $\mathcal{O}\big(\frac{d^2}{\sqrt{T}}\big)$ $\big[\mathcal{O}\big(d\sqrt{\frac{\ln(T)}{T}}\big)$ Agarwal et al. (2010) $\big]$ |
| Two-point | $\mathcal{F}_{\mathsf{lip}}$: $\mathcal{O}\big(\sqrt{\frac{d}{T}}\big)$ $\big[\mathcal{O}\big(\sqrt{\frac{d}{T}}\big)$ Shamir (2017) $\big]$ | $\mathcal{F}_{\mathsf{lip}} \cap \mathcal{F}_{\mathsf{sc}}$: $\mathcal{O}\big(\frac{d\ln(T)}{T}\big)$ $\big[$ $\mathcal{O}\big(\frac{d^2 \ln(T)}{T}\big)$ Agarwal et al. (2010) $\big]$ |
| | $\mathcal{F}_{\mathsf{lip}} \cap \mathcal{F}_{\mathsf{smo}}$: $\mathcal{O}\big(\sqrt{\frac{d}{T}}\big)$ | $\mathcal{F}_{\mathsf{lip}} \cap \mathcal{F}_{\mathsf{smo}} \cap \mathcal{F}_{\mathsf{sc}}$: $\mathcal{O}\big(\frac{d\ln(T)}{T}\big)$ $\big[\mathcal{O}\big(\frac{d^2 \ln(T)}{T}\big)$ Agarwal et al. (2010) $\big]$ |

**Related Work.** Recently, many types of centralized zeroth-order optimization algorithms have been studied, and their convergence rates (and the way they depend on the dimension) have been established in different settings. For unconstrained convex optimization, Nesterov & Spokoiny (2017) develops several types of two-point gradient estimators and achieves convergence rates that scale with dimension as $\mathcal{O}(d^2)$. For constrained stochastic optimization, Duchi et al. (2015) establishes that the convergence rates are sharp up to factors at most logarithmic in the dimension. Zeroth-order optimization has a natural connection to bandit online optimization, where the latter focuses on dynamic environment where the objective functions are varying over time (see, e.g., Flaxman et al. (2005); Agarwal et al. (2010); Shamir (2013; 2017); Bubeck et al. (2017); Lattimore (2020); Hazan & Levy (2014)). In particular, the seminal work Flaxman et al. (2005) constructs a one-point gradient estimator (or one-point bandit feedback model) and achieves an $\mathcal{O}(d/T^{1/4})$ average regret. For two-point gradient estimator, Shamir (2017) establishes the tightness of the dimension-dependent factor $\mathcal{O}(\sqrt{d})$ in the framework of zeroth-order stochastic mirror descent.

It is worth zooming into the literature on distributed zeroth-order/bandit online optimization. Due to the absence of a central coordinator, the algorithms developed should always rely on local computations and communications (e.g., Yuan & Ho (2015); Yi et al. (2020); Jakovetic et al. (2018); Hajinezhad et al. (2019); Wang et al. (2019); Pang & Hu (2019); Hajinezhad & Zavlanos (2018); Wan et al. (2020)). This makes the convergence analysis of the distributed zeroth-order/bandit online optimization algorithms more challenging. In Yuan & Ho (2015), the authors develop a class of distributed zeroth-order optimization algorithms that require two functional evaluations at each iteration, and establishes asymptotic convergence of the algorithm. Non-asymptotic convergence is established in Jakovetic et al. (2018); Hajinezhad et al. (2019); Wang et al. (2019); Pang & Hu (2019); Hajinezhad & Zavlanos (2018), but the dimension-dependence factors are either $\mathcal{O}(d^2)$ or far from optimal. The work Yi et al. (2020) considers distributed online optimization with long-term constraints and establishes bounds on regret as well as constraint violations. To avoid Euclidean projection onto the constraint set, Wan et al. (2020) develops a distributed bandit online optimization algorithm based on conditional gradient descent and one-point bandit feedback, and achieves a regret scaling of $\mathcal{O}(T^{3/4}\sqrt{\ln T})$.

## 2 THE MAZOPA ALGORITHM AND ITS CONVERGENCE RATES

In this section, we present the MAZOPA algorithm and establish the convergence rates for the three function classes.

### 2.1 DISTRIBUTED ZEROTH-ORDER ORACLES

Let $\mathbf{n}$ be a random vector in $\mathbb{R}^d$ drawn from some probability distribution. Then

$$\hat{f}_i(\mathbf{x}; \delta) := \mathbb{E}_{\mathbf{n}} \left[ f_i(\mathbf{x} + \delta \mathbf{n}) \right] \tag{2}$$

is a *smoothed* function for $f_i$. Here $\delta > 0$ is a parameter setting the level of the smoothing. We introduce the following definition on distributed zeroth-order oracles (DistZOO).

**Definition 1** (DistZOO) *A vector $\tilde{\mathbf{g}}_i(\mathbf{x}; \delta) \in \mathbb{R}^d$ is called a distributed zeroth-order oracle at node $i$ if the following conditions hold:*

*(i) $\mathbb{E}\left[\tilde{\mathbf{g}}_i(\mathbf{x}; \delta)\right] = \nabla \hat{f}_i(\mathbf{x}; \delta)$ for all $\mathbf{x} \in \mathbb{R}^d$;*

*(ii) If $f_i \in \mathcal{F}_{\mathsf{lip}}(L_f)$, then $\hat{f}_i \in \mathcal{F}_{\mathsf{lip}}(L_f)$ as well, and there holds $\left| \hat{f}_i(\mathbf{x}; \delta) - f_i(\mathbf{x}) \right| \leq p_d L_f \delta$, with $p_d$ being some positive constant;*

*(iii) If $f_i \in \mathcal{F}_{\sf smo}(s_f)$, then $\left| \hat{f}_i(\mathbf{x}; \delta) - f_i(\mathbf{x}) \right| \leq \frac{1}{2} \tilde{p}_d s_f \delta^2$ with $\tilde{p}_d$ being some positive constant.*

A number of DistZOO satisfying Definition 1 can be obtained using existing gradient estimators, see, e.g., Liu et al. (2020). In the paper, we provide two representative gradient estimators that are commonly adopted in the literature. Let $\mathbf{u}_i$ be a random vector independently generated from a unit sphere $\mathbb{B}_1$ in $\mathbb{R}^d$. Then (e.g., Flaxman et al. (2005))

$$\tilde{\mathbf{g}}_i^{\sf OP}(\mathbf{x}; \delta) := f_i(\mathbf{x} + \delta_t \mathbf{u}_i) \mathbf{u}_i d / \delta \tag{3}$$

is a one-point DistZOO satisfying Definition 1. Moreover,

$$\tilde{\mathbf{g}}_i^{\sf TP}(\mathbf{x}; \delta) := \frac{d}{2\delta} \big( f_i(\mathbf{x} + \delta \mathbf{u}) - f_i(\mathbf{x} - \delta \mathbf{u}) \big) \mathbf{u} \tag{4}$$

is a two-point DistZOO satisfying Definition 1 (e.g., Shamir (2017)).

## 2.2 The MAZOPA Algorithm

We present the following Multi-Agent Zeroth-Order Projection Averaging (MAZOPA) algorithm, which consists of two steps, a local zeroth-order optimization step and a distributed averaging step. MAZOPA, whose pseudo-code is presented in Algorithm 1, is a variation of the multi-agent sub-gradient averaging algorithm proposed in Nedic et al. (2008); Nedic & Ozdaglar (2009); Nedic et al. (2010), where the local optimization step is executed by sub-gradient descent.

---

**Algorithm 1** MAZOPA: $\hat{\mathbf{x}}_i(T) = \text{MAZOPA}\,(\mathbf{x}_i(1), \eta_t, \delta_t, \mathcal{X})$

---

**Require:** step size $\eta_t$, DistZOO $\tilde{\mathbf{g}}_i(\mathbf{x}; \delta_t)$ with exploration parameter $\delta_t$ for all $i \in \mathcal{V}$
**Ensure:** $\mathbf{x}_i(1) \in \mathcal{X}, \forall i \in \mathcal{V}$
 1: **for** $t = 1$ to $T$ **do**
 2:     Node $i$ queries the DistZOO at point $\mathbf{x}_i(t)$ and receives $\tilde{\mathbf{g}}_i(\mathbf{x}_i(t); \delta_t)$
 3:     Node $i$ computes

$$\mathbf{v}_i(t) = \mathbf{x}_i(t) - \eta_t \cdot \tilde{\mathbf{g}}_i(\mathbf{x}_i(t); \delta_t)$$

 4:     Node $i$ updates its state by using the information received from its instant neighbors

$$\mathbf{x}_i(t + 1) = \mathbf{proj}_{\mathcal{X}} \Big( \sum_{j=1}^{N} [\mathbf{A}(t)]_{ij} \mathbf{v}_j(t) \Big)$$

 5: **end for**
**Output:** $\hat{\mathbf{x}}_i(T) = \frac{1}{T} \sum_{t=1}^{T} \mathbf{x}_i(t)$

---

## 2.3 Main Results

Let $\hat{\mathbf{x}}_i(T)$ be the output of Algorithm 1 at agent $i$. We denote the optimal solution of problem (1) by $\mathbf{x}^\star = \arg\min_{\mathbf{x} \in \mathcal{X}} f(\mathbf{x})$. Defining $\mathcal{X}^\circ := \{\mathbf{x} + \mathbf{u} \,:\, \mathbf{x} \in \mathcal{X}, \mathbf{u} \in \mathbb{B}_1\}$, we present the following results on the convergence rate of the MAZOPA algorithm.

**Theorem 1** *Let Assumption 1 hold. Let* DistZOO *take the form of* $\tilde{\mathbf{g}}_i^{\sf OP}(\cdot)$. *Further assume that* $|f_i(\mathbf{x}_i(t) + \delta_t \mathbf{u}_i(t))| \leq C$ *for all* $i \in \mathcal{V}$. *We have the following convergence results for every* $i \in \mathcal{V}$ *and all* $T \geq 1$.

 *(i) Consider $f_i \in \mathcal{F}_{\sf lip}(L_f, \mathcal{X}^\circ)$ for all $i \in \mathcal{V}$. Setting $\eta_t = \frac{1}{dT^{3/4}}$ and $\delta_t = \frac{1}{t^{1/4}}$, $t = 1, \ldots, T$, it holds that $\mathbb{E}\big[f(\hat{\mathbf{x}}_i(T))\big] - f(\mathbf{x}^\star) = \mathcal{O}\big(\frac{d}{T^{1/4}}\big)$.*

 *(ii) Consider $f_i \in \mathcal{F}_{\sf lip}(L_f, \mathcal{X}^\circ) \cap \mathcal{F}_{\sf smo}(s_f, \mathcal{X}^\circ)$. Setting $\eta_t = \frac{1}{dT^{2/3}}$ and $\delta_t = \frac{1}{t^{1/6}}$, $t = 1, \ldots, T$, it holds that $\mathbb{E}\big[f(\hat{\mathbf{x}}_i(T))\big] - f(\mathbf{x}^\star) = \mathcal{O}\big(\frac{d}{T^{1/3}}\big)$.*

**Theorem 2** *Let Assumption 1 hold. Let* DistZOO *take the form of* $\tilde{\mathbf{g}}_i^{\sf TP}(\cdot)$. *Set $\eta_t = \frac{1}{\sqrt{dT}}$ and $\delta_t = \frac{1}{\sqrt{t}}$, $t = 1, \ldots, T$. Consider $f_i \in \mathcal{F}_{\sf lip}(L_f, \mathcal{X}^\circ), i \in \mathcal{V}$. Then, for every $i \in \mathcal{V}$ and all $T \geq 1$, we have $\mathbb{E}\big[f(\hat{\mathbf{x}}_i(T))\big] - f(\mathbf{x}^\star) = \mathcal{O}\big(\sqrt{\frac{d}{T}}\big)$.*

With strong convexity, the convergence rates established above can be further strengthened.

**Theorem 3** *Let Assumption 1 hold. Let* DistZOO *take the form of* $\tilde{\mathbf{g}}_i^{\mathsf{OP}}(\cdot)$. *Further assume that* $|f_i(\mathbf{x}_i(t) + \delta_t \mathbf{u}_i(t))| \leq C$ *for all* $i \in \mathcal{V}$. *We have the following convergence results for every* $i \in \mathcal{V}$ *and all* $T \geq 1$.

(i) *Consider* $f_i \in \mathcal{F}_{\mathsf{lip}}(L_f, \mathcal{X}^\circ) \cap \mathcal{F}_{\mathsf{sc}}(\mu_f, \mathcal{X}^\circ)$ *for all* $i \in \mathcal{V}$. *Setting* $\eta_t = \frac{1}{\mu_f t}$ *and* $\delta_t = \frac{1}{t^{1/3}}$, $t = 1, \ldots, T$, *it holds that* $\mathbb{E}[f(\hat{\mathbf{x}}_i(T))] - f(\mathbf{x}^\star) = \mathcal{O}\big(\frac{d^2}{T^{1/3}}\big)$.

(ii) *Consider* $f_i \in \mathcal{F}_{\mathsf{lip}}(L_f, \mathcal{X}^\circ) \cap \mathcal{F}_{\mathsf{smo}}(s_f, \mathcal{X}^\circ) \cap \mathcal{F}_{\mathsf{sc}}(\mu_f, \mathcal{X}^\circ)$. *Setting* $\eta_t = \frac{1}{\mu_f t}$ *and* $\delta_t = \frac{1}{t^{1/4}}$, $t = 1, \ldots, T$, *it holds that* $\mathbb{E}[f(\hat{\mathbf{x}}_i(T))] - f(\mathbf{x}^\star) = \mathcal{O}\big(\frac{d^2}{\sqrt{T}}\big)$.

**Theorem 4** *Let Assumption 1 hold. Let* DistZOO *take the form of* $\tilde{\mathbf{g}}_i^{\mathsf{TP}}(\cdot)$. *Set* $\eta_t = \frac{1}{\mu_f t}$ *and* $\delta_t = \frac{1}{t}$, $t = 1, \ldots, T$. *Consider* $f_i \in \mathcal{F}_{\mathsf{lip}}(L_f, \mathcal{X}^\circ) \cap \mathcal{F}_{\mathsf{sc}}(\mu_f, \mathcal{X}^\circ), i \in \mathcal{V}$. *Then, for every* $i \in \mathcal{V}$ *and all* $T \geq 1$, *we have* $\mathbb{E}[f(\hat{\mathbf{x}}_i(T))] - f(\mathbf{x}^\star) = \mathcal{O}\big(\frac{d \ln(T)}{T}\big)$.

## 3    MULTISTAGE MAZOPA: ADAPTIVE LOCAL DESCENT

We now propose a multi-stage variant of Algorithm 1. We impose the following compactness assumption on the constraint set $\mathcal{X}$.

**Assumption 2** *There exists* $0 < R_{\mathcal{X}} < \infty$ *such that* $\|\mathbf{x}\| \leq R_{\mathcal{X}}$ *for all* $\mathbf{x} \in \mathcal{X}$.

### 3.1    THE ALGORITHM

The basic idea is to divide the optimization process into a sequence of epochs, each of which has an exponentially decreasing step size and an exponentially increasing iteration number. The updates in the inner loop of each stage are just made according to Algorithm 1 with fixed step size. In each stage only the average point is maintained and used as the starting point of the next stage. This idea of setting up multi-stage optimization algorithms was originally explored in Hazan & Kale (2011).

Take positive integers $m \geq 1$ and $a \geq 2$. Let $k^\natural = \lfloor \log_a \big( \frac{T}{m} + 1 \big) \rfloor$, where $\lfloor x \rfloor$ represents the largest inter with value no greater than $x \in \mathbb{R}$. We divide the $T$ time steps into $k^\natural$ epochs by

$$\begin{aligned} \text{Epoch } 1 : & \quad 1, \ldots, T^{(1)}; \\ \text{Epoch } 2 : & \quad T^{(1)} + 1, \ldots, T^{(2)}; \\ & \quad \vdots \\ \text{Epoch } k^\natural : & \quad T^{(k^\natural - 1)} + 1, \ldots, T^{(k^\natural)}. \end{aligned}$$

Here $T^{(1)} = m, T^{(2)} = am, \ldots, T^{(k^\natural)} = a^{k^\natural} m$. For the $j$th epoch, all agents will run the MAZOPA algorithm, and denote output of the $j$-th epoch at agent $i$ by $\hat{\mathbf{x}}_i^{(j)}(T^{(j)})$. The pseudo-code of the resulting multi-stage MAZOPA is presented in Algorithm 2.

Compared to the MAZOPA algorithm, the multi-stage MAZOPA has the following advantages:

(i) Multistage MAZOPA only requires each node projects its estimates onto the ball $\mathbb{B}_{R_{\mathcal{X}}}$, rather than the constraint set $\mathcal{X}$ in each epoch. In particular, multistage MAZOPA algorithm significantly reduces the number of Euclidean projections onto the constraint set $\mathcal{X}$ from $T$ to $k^\natural$. This makes the algorithm more computationally efficient.

(ii) Multistage MAZOPA better utilizes the step size rules, in the sense that at earlier epochs of the algorithm, larger step sizes are adopted to facilitate convergence, while smaller step sizes are adopted to achieve better accuracy at later epochs.

### 3.2    OPTIMAL CONVERGENCE RATES

We now modify the definitions of $\mathcal{F}_{\mathsf{lip}}$, $\mathcal{F}_{\mathsf{smo}}$ and $\mathcal{F}_{\mathsf{sc}}$ by replacing $\mathcal{X}$ with $\mathbb{B}_{R_{\mathcal{X}}}$, with slight abuse of notation. As it turns out, the multistage MAZOPA enjoys refined convergence rates.

---

**Algorithm 2** Multistage MAZOPA

---

**Require:** exploration parameter $\delta^{(1)}$, step size $\eta^{(1)}$, $T^{(1)} = m$, total number of iterations $T$, integer $a \geq 2$, and scalar $b > 1$

**Ensure:** $\mathbf{x}_i^{(1)}(1) \in \mathcal{X}$ for all $i \in \mathcal{V}$, and set $k = 1$

1: **while** $j = 1, \ldots, k^{\natural}$ **do**
2:   Call Algorithm 1 to obtain

$$\hat{\mathbf{x}}_i^{(j)}(T^{(j)}) = \mathsf{MAZOPA}\left(\mathbf{x}_i^{(j)}(1), \eta^{(j)}, \delta^{(j)}, \mathbb{B}_{R_{\mathcal{X}}}\right)$$

3:   Compute $\mathbf{x}_i^{(j+1)}(1) = \mathbf{proj}_{\mathcal{X}}\left(\hat{\mathbf{x}}_i^{(j)}(T^{(j)})\right)$
4:   Update $\eta^{(k+1)} = \frac{1}{a}\eta^{(k)}$ and $\delta^{(j+1)} = \frac{1}{b}\delta^{(j)}$
5:   Update $T^{(j+1)} = aT^{(j)}$
6:   Update $j = j + 1$
7: **end while**

**Output:** $\bar{\mathbf{x}}_i(T) = \mathbf{proj}_{\mathcal{X}}\left(\hat{\mathbf{x}}_i^{(k^{\natural})}(T^{(k^{\natural})})\right)$

---

**Theorem 5** *Let Assumptions 1 and 2 hold. Let* $\mathsf{DistZOO}$ *take the form of* $\tilde{\mathbf{g}}_i^{\mathsf{TP}}(\cdot)$. *Set* $a = b$, $T^{(1)} = m = 1$, $\eta^{(1)} = \frac{4a}{3\mu_f}$ *and* $\delta^{(1)} = 1$. *Consider* $f_i \in \mathcal{F}_{\mathsf{lip}}(L_f, \mathbb{B}_{R_{\mathcal{X}}}) \cap \mathcal{F}_{\mathsf{sc}}(\mu_f, \mathbb{B}_{R_{\mathcal{X}}})$, $i \in \mathcal{V}$. *We have* $\mathbb{E}\left[\max_{i \in \mathcal{V}}\left\{\|\bar{\mathbf{x}}_i(T) - \mathbf{x}^{\star}\|^2\right\}\right] = \mathcal{O}\left(\frac{d}{T+1}\right)$.

The idea of the analysis leading to Theorem 5 can also be extended to one-point oracles. If $\mathsf{DistZOO}$ takes the form of $\tilde{\mathbf{g}}_i^{\mathsf{OP}}(\cdot)$, one needs to impose the following assumption on the objective functions, that is, $|f_i(\mathbf{x}_i(t) + \delta_t \mathbf{u}_i(t))| \leq C$ for all $i \in \mathcal{V}$. For $f_i \in \mathcal{F}_{\mathsf{lip}}(L_f, \mathbb{B}_{R_{\mathcal{X}}}) \cap \mathcal{F}_{\mathsf{sc}}(\mu_f, \mathbb{B}_{R_{\mathcal{X}}})$, setting $a = b^3$, the final estimates enjoy a convergence rate of $\mathbb{E}\left[\max_{i \in \mathcal{V}}\left\{\|\bar{\mathbf{x}}_i(T) - \mathbf{x}^{\star}\|^2\right\}\right] = \mathcal{O}\left(\frac{d^2}{(T+1)^{1/3}}\right)$. For $f_i \in \mathcal{F}_{\mathsf{lip}}(L_f, \mathbb{B}_{R_{\mathcal{X}}}) \cap \mathcal{F}_{\mathsf{smo}}(s_f, \mathbb{B}_{R_{\mathcal{X}}}) \cap \mathcal{F}_{\mathsf{sc}}(\mu_f, \mathbb{B}_{R_{\mathcal{X}}})$, setting $a = b^4$, there holds $\mathbb{E}\left[\max_{i \in \mathcal{V}}\left\{\|\bar{\mathbf{x}}_i(T) - \mathbf{x}^{\star}\|^2\right\}\right] = \mathcal{O}\left(\frac{d^2}{\sqrt{T+1}}\right)$.

## 4 NUMERICAL EXAMPLES

In this section, we evaluate the performance of the proposed algorithms on a distributed ridge regression problem.

**System setup**. The optimization problem has the following form:

$$\begin{aligned} \text{minimize} \quad & f(\mathbf{x}) = \sum_{i=1}^{N}\left(\tfrac{1}{2}(\mathbf{a}_i^{\mathsf{T}}\mathbf{x} - b_i)^2 + \rho\|\mathbf{x}\|^2\right) \\ \text{subject to} \quad & \|\mathbf{x}\|_1 \leq k \end{aligned} \tag{5}$$

where $\mathbf{x} \in \mathbb{R}^d$ is the optimization variable, the data pair $(\mathbf{a}_i, b_i) \in \mathbb{R}^d \times \mathbb{R}$ is only known to node $i$ with $\mathbf{a}_i$ and $b_i$ being generated uniformly from the unit normal distribution.

**Network setup**. We implement the proposed algorithms over a randomly generated network that consists of $N = 50$ nodes, which is shown in Fig. 1. In the simulations, we set $d = 10$, $k = 3/4$, $\rho = 1/2$, and $R_{\mathcal{W}} = 3/4$. We evaluate the performance of the algorithms via the average of 10 implementations. The weight matrix associated with the graph is generated according to the maximum-degree weights:

$$[\mathbf{A}(t)]_{ij} = \begin{cases} \frac{1}{1 + d_{\max}}, & (j, i) \in \mathcal{E}_t \\ 1 - \frac{d_i}{1 + d_{\max}}, & i = j \\ 0, & (j, i) \notin \mathcal{E}_t \end{cases}$$

where $d_{\max} = \max_{i \in \mathcal{V}}\{d_i\}$ is the maximum degree of $\mathcal{G}_t$ ($d_i$ denotes the degree of node $i$).

**Results**. The performance of algorithms MAZOPA and multistage MAZOPA is illustrated via plotting the maximum function errors, $\max_{i \in \mathcal{V}} f(\hat{x}_i(T))$ and $\max_{i \in \mathcal{V}} f(\bar{x}_i(T))$, as a function of the number of iterations $T$ in Fig. 2. As a benchmark, the convergence performance of the gradient

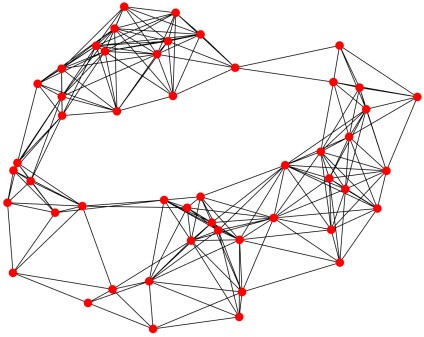

Fig. 1. A random network of 50 nodes.

algorithm is displayed in Fig. 2 as well. From the numerical results it is clear that the maximum function errors are vanishing for all zerroth-order algorithms. In fact, the convergence performance of two-point MAZOPA is even comparable to the gradient method. Moreover, the multistage variants in general exhibit better convergence performance, and this is more obvious for the case of two-point MAZOPA. These numerical results are in compliance with the theoretical findings in the paper.

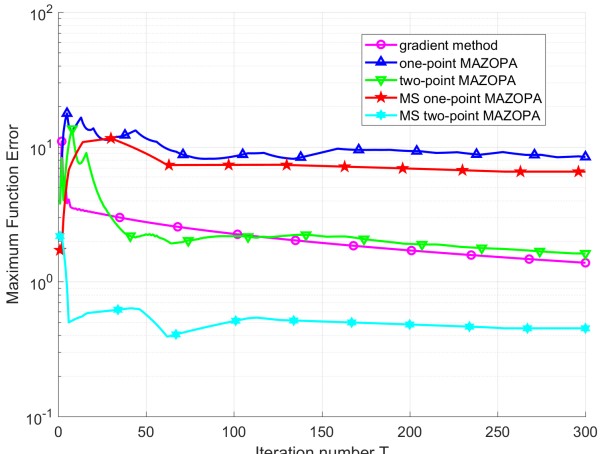

Fig. 2. Convergence performance of the algorithms.

**Reproduction of the results.** The code used for producing this numerical example is provided in the suplementary material.

## 5    CONCLUSIONS

We have established a series of convergence rates for distributed zeroth-order subgradient algorithms that match their centralized counterpart for Lipschitz, smooth, and strongly convex function classes. These results provided the theoretical benchmarks for zeroth-order approaches over complex dynamic networks. We also proposed a multi-stage variant of the algorithm that better utilizes the learning rates and attains even improved convergence rates. In future work, it is worth exploring the connection between the convergence rates and the underlying communication complexity for distributed zeroth-order algorithms.

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

## A  KEY LEMMAS

We first establish the basic convergence result for Algorithm 1 that is based on DistZOO $\tilde{\mathbf{g}}_i(\mathbf{x}; \delta_t)$, which plays a crucial role in subsequent analyses. We will sometimes use $i^\bullet$ to denote a node in $\mathcal{V}$ just to highlight the focus on a given node (but $i^\bullet$ indeed may take any value in $\mathcal{V}$ and therefore it is a generic node).

**Lemma 1** *Let Assumption 1 hold. Let $\tilde{\mathbf{g}}_i(\mathbf{x}_i(t); \delta_t)$ be a* DistZOO *that satisfies Definition 1 (i) and (ii). Then, for any $i^\bullet \in \mathcal{V}$ and all $T \geq 1$, there holds*

$$\sum_{t=1}^{T} \left( \mathbb{E}\left[ f(\mathbf{x}_{i^\bullet}(t)) \right] - f(\mathbf{x}^\star) \right) \leq \sum_{t=1}^{T} \sum_{i=1}^{N} \mathbb{E}\left[ \left| f_i(\mathbf{x}^\star) - \hat{f}_i(\mathbf{x}^\star; \delta_t) \right| \right]$$

$$+ \sum_{t=1}^{T} \sum_{i=1}^{N} \mathbb{E}\left[ \left| f_i(\mathbf{x}_{i^\bullet}(t)) - \hat{f}_i(\mathbf{x}_{i^\bullet}(t); \delta_t) \right| \right]$$

$$+ \sum_{t=1}^{T} \frac{\mathbb{E}\left[ \Lambda^\star(t) \right] - \mathbb{E}\left[ \Lambda^\star(t+1) \right]}{2\eta_t} + p_1 L_f$$

$$+ \frac{1}{2} \sum_{t=1}^{T} \eta_t \sum_{i=1}^{N} \mathbb{E}\left[ \| \tilde{\mathbf{g}}_i(\mathbf{x}_i(t); \delta_t) \|^2 \right]$$

$$+ p_2 L_f \sum_{t=1}^{T-1} \eta_t \sum_{i=1}^{N} \mathbb{E}\left[ \| \tilde{\mathbf{g}}_i(\mathbf{x}_i(t); \delta_t) \| \right]$$

*where $p_1 = 2N \max_{i \in \mathcal{V}} \{ \| \mathbf{x}_{\mathrm{avg}}(1) - \mathbf{x}_i(1) \| \} + \frac{2N\alpha\beta}{1-\beta} \left( \sum_{i=1}^{N} \| \mathbf{x}_i(1) \| \right)$, $p_2 = 2N \left( \frac{3\alpha}{1-\beta} + 4 \right)$, and $\Lambda^\star(t) = \sum_{i=1}^{N} \| \mathbf{x}_i(t) - \mathbf{x}^\star \|^2$ with $\mathbf{x}_{\mathrm{avg}}(1) = \frac{1}{N} \sum_{i=1}^{N} \mathbf{x}_i(1)$, $\alpha = \left( 1 - \frac{\xi}{4N^2} \right)^{-2}$, and $\beta = \left( 1 - \frac{\xi}{4N^2} \right)^{1/B}$.*

Before presenting the proof of Lemma 1, we provide the following two supporting lemmas. The first lemma characterizes the convergence property of the transition matrix induced by weight matrix $\mathbf{A}(t)$ (see Nedic et al. (2008)).

**Lemma 2** *Define the transition matrix as $\mathbf{A}(t : \ell) = \mathbf{A}(t)\mathbf{A}(t-1) \cdots \mathbf{A}(\ell+1)\mathbf{A}(\ell)$ for all $t \geq \ell \geq 1$, and write $\mathbf{A}(t : t) = \mathbf{A}(t)$. $\mathbf{A}(t : \ell)$ satisfies*

$$\left| [\mathbf{A}(t : \ell)]_{ij} - \frac{1}{N} \right| \leq \alpha \beta^{t-\ell+1}$$

*where $\alpha = \left( 1 - \frac{\xi}{4N^2} \right)^{-2}$ and $\beta = \left( 1 - \frac{\xi}{4N^2} \right)^{1/B}$.*

The second lemma establishes the accumulated disagreement for every node in the network.

**Lemma 3 (Disagreement)** *Let Assumption 1 hold. For every node $i \in \mathcal{V}$, we have*

$$\sum_{t=1}^{T} \| \mathbf{x}_{\mathrm{avg}}(t) - \mathbf{x}_i(t) \| \leq \| \mathbf{x}_{\mathrm{avg}}(1) - \mathbf{x}_i(1) \| + \frac{\alpha\beta}{1-\beta} \left( \sum_{i=1}^{N} \| \mathbf{x}_i(1) \| \right)$$

$$+ \left( \frac{3\alpha}{1-\beta} + 4 \right) \sum_{t=1}^{T-1} \eta_t \sum_{i=1}^{N} \| \tilde{\mathbf{g}}_i(\mathbf{x}_i(t); \delta_t) \|$$

*where $\mathbf{x}_{\mathrm{avg}}(t) = \frac{1}{N} \sum_{i=1}^{N} \mathbf{x}_i(t)$.*

*Proof.* To simplify the presentation, we denote

$$\tilde{\mathbf{v}}_i(t) = \sum_{j=1}^{N} [\mathbf{A}(t)]_{ij} \mathbf{v}_j(t)$$

$$\mathbf{s}_i(t) = \mathbf{proj}_{\mathcal{X}}(\tilde{\mathbf{v}}_i(t)) - \tilde{\mathbf{v}}_i(t).$$

Step 3 in Algorithm 1 can be rewritten as

$$\mathbf{x}_i(t+1) = \tilde{\mathbf{v}}_i(t) + \mathbf{s}_i(t).$$

Our analysis relies on the estimate of $\|\mathbf{s}_i(t)\|$, which can be bounded as follows:

$$\|\mathbf{s}_i(t)\| \leq \left\|\mathbf{proj}_{\mathcal{X}}\left(\tilde{\mathbf{v}}_i(t)\right) - \sum_{j=1}^{N}[\mathbf{A}(t)]_{ij}\mathbf{x}_j(t)\right\| + \sum_{j=1}^{N}[\mathbf{A}(t)]_{ij}\left\|\eta_t\tilde{\mathbf{g}}_j(\mathbf{x}_j(t);\delta_t)\right\|$$

where the inequality is based on Step 3 in Algorithm 1 and $\mathbf{A}(t)$ is double stochastic (cf. Assumption 1). Using the non-expansiveness of the Euclidean projection $\mathbf{proj}_{\mathcal{X}}(\cdot)$ and the fact that $\sum_{j=1}^{N}[\mathbf{A}(t)]_{ij}\mathbf{x}_j(t) \in \mathcal{X}$, we have

$$\|\mathbf{s}_i(t)\| \leq 2\sum_{j=1}^{N}[\mathbf{A}(t)]_{ij}\left\|\eta_t\tilde{\mathbf{g}}_j(\mathbf{x}_j(t);\delta_t)\right\|. \tag{6}$$

We now derive the general expressions for $\mathbf{x}_{\mathrm{avg}}(t+1)$ and $\mathbf{x}_i(t+1)$, respectively. For $\mathbf{x}_{\mathrm{avg}}(t+1)$, we have

$$\mathbf{x}_{\mathrm{avg}}(t+1) = \mathbf{x}_{\mathrm{avg}}(t) - \frac{1}{N}\sum_{i=1}^{N}\eta_t\tilde{\mathbf{g}}_i(\mathbf{x}_i(t);\delta_t) + \frac{1}{N}\sum_{i=1}^{N}\mathbf{s}_i(t).$$

Applying the preceding inequality recursively, we get

$$\mathbf{x}_{\mathrm{avg}}(t+1) = \mathbf{x}_{\mathrm{avg}}(1) - \sum_{\ell=1}^{t}\frac{1}{N}\sum_{i=1}^{N}\eta_\ell\tilde{\mathbf{g}}_i(\mathbf{x}_i(\ell);\delta_\ell) + \sum_{\ell=1}^{t}\frac{1}{N}\sum_{i=1}^{N}\mathbf{s}_i(\ell). \tag{7}$$

Similarly, for $\mathbf{x}_i(t+1)$, we have

$$\mathbf{x}_i(t+1) = \sum_{j=1}^{N}[\mathbf{A}(t:1)]_{ij}\mathbf{x}_j(1) - \sum_{\ell=1}^{t}\sum_{j=1}^{N}[\mathbf{A}(t:\ell)]_{ij}\eta_\ell\tilde{\mathbf{g}}_j(\mathbf{x}_j(\ell);\delta_\ell) + \sum_{\ell=1}^{t-1}\sum_{j=1}^{N}[\mathbf{A}(t:\ell+1)]_{ij}\mathbf{s}_j(\ell) + \mathbf{s}_i(t). \tag{8}$$

Combining (7) and (8), gives

$$\|\mathbf{x}_{\mathrm{avg}}(t+1) - \mathbf{x}_i(t+1)\| \leq \sum_{j=1}^{N}\left|[\mathbf{A}(t:1)]_{ij} - \frac{1}{N}\right|\|\mathbf{x}_j(1)\| + \sum_{\ell=1}^{t}\sum_{j=1}^{N}\left|[\mathbf{A}(t:\ell)]_{ij} - \frac{1}{N}\right|\eta_\ell\|\tilde{\mathbf{g}}_j(\mathbf{x}_j(\ell);\delta_\ell)\|$$

$$+ \sum_{\ell=1}^{t-1}\sum_{j=1}^{N}\left|[\mathbf{A}(t:\ell+1)]_{ij} - \frac{1}{N}\right|\|\mathbf{s}_j(\ell)\| + \|\mathbf{s}_i(t)\| + \frac{1}{N}\sum_{i=1}^{N}\|\mathbf{s}_i(t)\|. \tag{9}$$

Combining the results in (6), (9) and Lemma 2, leads to

$$\|\mathbf{x}_{\mathrm{avg}}(t+1) - \mathbf{x}_i(t+1)\| \leq \alpha\beta^t\left(\sum_{i=1}^{N}\|\mathbf{x}_i(1)\|\right) + 3\alpha\sum_{\ell=1}^{t}\beta^{t-\ell}\eta_\ell\sum_{i=1}^{N}\|\tilde{\mathbf{g}}_i(\mathbf{x}_i(\ell);\delta_\ell)\| + 4\eta_t\sum_{i=1}^{N}\|\tilde{\mathbf{g}}_i(\mathbf{x}_i(t);\delta_t)\|$$

where we used the following relation, based on (6):

$$\|\mathbf{s}_i(t)\| + \frac{1}{N}\sum_{i=1}^{N}\|\mathbf{s}_i(t)\| \leq 2\sum_{i=1}^{N}\|\mathbf{s}_i(t)\| \leq 4\sum_{i=1}^{N}\sum_{j=1}^{N}[\mathbf{A}(t)]_{ij}\|\eta_t\tilde{\mathbf{g}}_j(\mathbf{x}_j(t);\delta_t)\| \leq 4\eta_t\sum_{i=1}^{N}\|\tilde{\mathbf{g}}_i(\mathbf{x}_i(t);\delta_t)\|.$$

This implies that

$$\sum_{t=1}^{T}\|\mathbf{x}_{\mathrm{avg}}(t) - \mathbf{x}_i(t)\| \leq \|\mathbf{x}_{\mathrm{avg}}(1) - \mathbf{x}_i(1)\| + \alpha\left(\sum_{i=1}^{N}\|\mathbf{x}_i(1)\|\right)\sum_{t=1}^{T-1}\beta^t$$

$$+ 3\alpha\sum_{t=1}^{T-1}\sum_{\ell=1}^{t}\beta^{t-\ell}\eta_\ell\sum_{i=1}^{N}\|\tilde{\mathbf{g}}_i(\mathbf{x}_i(\ell);\delta_\ell)\| + 4\sum_{t=1}^{T-1}\eta_t\sum_{i=1}^{N}\|\tilde{\mathbf{g}}_i(\mathbf{x}_i(t);\delta_t)\|. \tag{10}$$

This, in combination with (10), leads to the final bound. □

[*Proof of Lemma 1*]. Denote

$$\Lambda(t) = \sum_{i=1}^{N} \|\mathbf{x}_i(t) - \mathbf{x}\|^2, \qquad \forall \mathbf{x} \in \mathfrak{X}, t \geq 1. \tag{11}$$

We follow the standard analysis by deriving the general evolution of $\Delta(t)$,

$$\Lambda(t+1) = \sum_{i=1}^{N} \left\| \mathbf{proj}_{\mathfrak{X}} \left( \sum_{j=1}^{N} [\mathbf{A}(t)]_{ij} \mathbf{v}_j(t) \right) - \mathbf{x} \right\|^2 \leq \sum_{i=1}^{N} \|\mathbf{v}_i(t) - \mathbf{x}\|^2 \tag{12}$$

where the inequality follows from the non-expansiveness of the Euclidean projection and the convexity of norm square function. Expanding the term further gives

$$\Lambda(t+1) = \Lambda(t) + \sum_{i=1}^{N} \|\eta_t \tilde{\mathbf{g}}_i(\mathbf{x}_i(t); \delta_t)\|^2 - 2\eta_t \sum_{i=1}^{N} \langle \tilde{\mathbf{g}}_i(\mathbf{x}_i(t); \delta_t), \mathbf{x}_i(t) - \mathbf{x} \rangle \tag{13}$$

Taking the expectation on both sides and using the following property of DistZOO (cf. Definition 1(i)):

$$\mathbb{E}[\tilde{\mathbf{g}}_i(\mathbf{x}_i(t); \delta_t)] = \nabla \hat{f}_i(\mathbf{x}_i(t); \delta_t)$$

we further obtain

$$\mathbb{E}[\Lambda(t+1)] = \mathbb{E}[\Lambda(t)] + \eta_t^2 \sum_{i=1}^{N} \mathbb{E}[\|\tilde{\mathbf{g}}_i(\mathbf{x}_i(t); \delta_t)\|^2] - 2\eta_t \sum_{i=1}^{N} \left( \mathbb{E}[\hat{f}_i(\mathbf{x}_i(t); \delta_t)] - \hat{f}_i(\mathbf{x}; \delta_t) \right) \tag{14}$$

which implies

$$\sum_{t=1}^{T} \sum_{i=1}^{N} \mathbb{E}[\hat{f}_i(\mathbf{x}_i(t); \delta_t)] - \hat{f}(\mathbf{x}; \delta_t) \leq \sum_{t=1}^{T} \frac{\mathbb{E}[\Lambda(t)] - \mathbb{E}[\Lambda(t+1)]}{2\eta_t} + \frac{1}{2} \sum_{t=1}^{T} \eta_t \sum_{i=1}^{N} \mathbb{E}[\|\tilde{\mathbf{g}}_i(\mathbf{x}_i(t); \delta_t)\|^2]. \tag{15}$$

We turn our attention to the left-hand side of (15). By adding and subtracting the term $\hat{f}_i(\mathbf{x}_{i\bullet}(t); \delta_t)$ and using the Lipschitz continuity of $\hat{f}_i$ (cf. Definition 1(ii)), it follows that

$$\sum_{t=1}^{T} \sum_{i=1}^{N} \hat{f}_i(\mathbf{x}_i(t); \delta_t) \geq \sum_{t=1}^{T} \hat{f}(\mathbf{x}_{i\bullet}(t); \delta_t) - L_f \sum_{t=1}^{T} \sum_{i=1}^{N} \|\mathbf{x}_i(t) - \mathbf{x}_{i\bullet}(t)\|$$

which, together with (15), yields

$$\sum_{t=1}^{T} \mathbb{E}[\hat{f}(\mathbf{x}_{i\bullet}(t); \delta_t)] - \hat{f}(\mathbf{x}; \delta_t) \leq \sum_{t=1}^{T} \frac{\mathbb{E}[\Lambda(t)] - \mathbb{E}[\Lambda(t+1)]}{2\eta_t}$$

$$+ \frac{1}{2} \sum_{t=1}^{T} \eta_t \sum_{i=1}^{N} \mathbb{E}[\|\tilde{\mathbf{g}}_i(\mathbf{x}_i(t); \delta_t)\|^2] + L_f \sum_{t=1}^{T} \sum_{i=1}^{N} \|\mathbf{x}_i(t) - \mathbf{x}_{i\bullet}(t)\|. \tag{16}$$

The desired result follows by relating the left-hand side to the original function $f$, using the disagreement estimate in Lemma 3, and setting $\mathbf{x} = \mathbf{x}^\star$. □

## B   PROOFS OF THEOREMS 1 AND 2

We first provide the the following lemma that characterizes the properties of the DistZOOs in (3) and (4). Its proof can be derived by resorting to Flaxman et al. (2005); Shamir (2017), which we omit here to save space.

**Lemma 4** *Suppose that $f_i \in \mathcal{F}_{\mathsf{lip}}(L_f)$ for all $i \in \mathcal{V}$. We have the following.*

*(i) For $\tilde{\mathbf{g}}_i^{\mathsf{OP}}(\cdot)$, there hold $p_d = 1$ and*

$$\mathbb{E}\big[\|\tilde{\mathbf{g}}_i^{\mathsf{OP}}(\mathbf{x}_i(t); \delta_t)\|^2\big] \leq \left(\frac{Cd}{\delta_t}\right)^2$$

*where $C = \max_{i \in \mathcal{V}} |f_i(\mathbf{x}_i(t) + \delta_t \mathbf{u}_i(t))|$ with $\mathbf{x}_i(t) \in \mathcal{X}$ and $\mathbf{u}_i(t)$ uniformly drawn from $\mathbb{B}_1$. We have $\tilde{p}_d = 1$ when $f_i \in \mathcal{F}_{\mathsf{smo}}(s_f)$.*

*(ii) For $\tilde{\mathbf{g}}_i^{\mathsf{TP}}(\cdot)$, there hold $p_d = 1$ and*

$$\mathbb{E}\big[\|\tilde{\mathbf{g}}_i^{\mathsf{TP}}(\mathbf{x}_i(t); \delta_t)\|^2\big] \leq cL_f^2 d$$

*where $c$ is some universal constant. In addition, $\tilde{p}_d = 1$ when $f_i \in \mathcal{F}_{\mathsf{smo}}(s_f)$.*

Now we are ready to prove Theorems 1 and 2.

(i) First, using the property of the DistZOO $\tilde{\mathbf{g}}_i^{\mathsf{OP}}(\cdot)$ (cf. Definition 1(iii)), it follows that

$$\sum_{i=1}^{N} \big|f_i(\mathbf{x}_{i\bullet}(t)) - \hat{f}_i(\mathbf{x}_{i\bullet}(t); \delta_t)\big| \leq NL_f \delta_t$$

$$\sum_{i=1}^{N} \big|f_i(\mathbf{x}^\star) - \hat{f}_i(\mathbf{x}^\star; \delta_t)\big| \leq NL_f \delta_t \tag{17}$$

We now focus on the case of $f_i \in \mathcal{F}_{\mathsf{lip}}(L_f, \mathcal{X}^\circ)$. Combining with inequality (17), the results in Lemmas 4(i) and 1, gives

$$\sum_{t=1}^{T} \big(\mathbb{E}[f(\mathbf{x}_{i\bullet}(t))] - f(\mathbf{x}^\star)\big) \leq p_1 L_f + 2NL_f \sum_{t=1}^{T} \delta_t + \frac{1}{2\eta_t} \mathbb{E}[\Lambda^\star(1)]$$

$$+ \frac{1}{2} NC^2 d^2 \sum_{t=1}^{T} \frac{\eta_t}{\delta_t^2} + p_2 NL_f Cd \sum_{t=1}^{T} \frac{\eta_t}{\delta_t} \tag{18}$$

where we used the fact that $\eta_t$ is a function of $T$ and Jensen's inequality, i.e.,

$$\mathbb{E}\big[\|\tilde{\mathbf{g}}_i^{\mathsf{OP}}(\mathbf{x}_i(t); \delta_t)\|\big] \leq \Big(\mathbb{E}\big[\|\tilde{\mathbf{g}}_i^{\mathsf{OP}}(\mathbf{x}_i(t); \delta_t)\|^2\big]\Big)^{1/2} \leq \frac{Cd}{\delta_t}.$$

Substituting the explicit expressions of $\eta = \frac{1}{dT^{3/4}}$ and $\delta_t = \frac{1}{t^{1/4}}$ into (18) and dividing both sides by $T$, we find that

$$\frac{1}{T} \sum_{t=1}^{T} \big(\mathbb{E}[f(\mathbf{x}_{i\bullet}(t))] - f(\mathbf{x}^\star)\big) = \mathcal{O}\left(\frac{d}{T^{1/4}}\right) \tag{19}$$

where we used the following inequality thet $\sum_{t=1}^{T} t^a = \mathcal{O}(T^{1+a}), \forall a \neq -1$. The desired result follows by using the convexity of function $f$, i.e., $\frac{1}{T} \sum_{t=1}^{T} f(\mathbf{x}_{i\bullet}(t)) \geq f(\hat{\mathbf{x}}_{i\bullet}(T))$.

When $f_i \in \mathcal{F}_{\mathsf{lip}}(L_f, \mathcal{X}^\circ) \cap \mathcal{F}_{\mathsf{smo}}(s_f, \mathcal{X}^\circ)$, it follows from the property of the DistZOO $\tilde{\mathbf{g}}_i^{\mathsf{OP}}(\cdot)$ (cf. Definition 1(iii)) that

$$\sum_{i=1}^{N} \big|f_i(\mathbf{x}_{i\bullet}(t)) - \hat{f}_i(\mathbf{x}_{i\bullet}(t); \delta_t)\big| \leq \frac{1}{2} N s_f \delta_t^2$$

$$\sum_{i=1}^{N} \big|f_i(\mathbf{x}^\star) - \hat{f}_i(\mathbf{x}^\star; \delta_t)\big| \leq \frac{1}{2} N s_f \delta_t^2 . \tag{20}$$

We then combine the preceding inequality and the results in Lemmas 4(i) and 1 to get

$$\sum_{t=1}^{T} \big(\mathbb{E}[f(\mathbf{x}_{i\bullet}(t))] - f(\mathbf{x}^\star)\big) \leq p_1 L_f + N s_f \sum_{t=1}^{T} \delta_t^2 + \frac{1}{2\eta_t} \mathbb{E}[\Lambda^\star(1)]$$

$$+ \frac{1}{2} NC^2 d^2 \sum_{t=1}^{T} \frac{\eta_t}{\delta_t^2} + p_2 NL_f Cd \sum_{t=1}^{T} \frac{\eta_t}{\delta_t}. \tag{21}$$

In contrast with the bound in (18), the second term on the right-hand side of (21) now becomes $Ns_f \sum_{t=1}^{T} \delta_t^2$, which gives us much more space when choosing $\delta_t$; we can show that the choices of $\eta = \frac{1}{dT^{2/3}}$ and $\delta_t = \frac{1}{t^{1/6}}$ yield the optimal convergence rate $\mathcal{O}\left(\frac{d}{T^{1/3}}\right)$.

(ii) When $f_i \in \mathcal{F}_{\mathsf{lip}}(L_f, \mathcal{X}^\circ)$, we have the following result for Algorithm 1 running with DistZOO $\tilde{\mathbf{g}}_i^{\mathsf{TP}}(\cdot)$:

$$\sum_{t=1}^{T} \left(\mathbb{E}\left[f(\mathbf{x}_{i\bullet}(t))\right] - f(\mathbf{x}^\star)\right) \leq p_1 L_f + 2NL_f \sum_{t=1}^{T} \delta_t + \frac{1}{2\eta_t} \mathbb{E}\left[\Lambda^\star(1)\right]$$
$$+ \left(\frac{1}{2}cd + p_2\sqrt{c}\sqrt{d}\right) NL_f^2 \eta_t T \tag{22}$$

where we have used the bounds in (17), Lemma 4(ii) and Lemma 1. Then we can deduce from the terms $\frac{1}{\eta_t}$ and $\eta_t T$ that the optimal choice of $\eta_t$ is $\frac{1}{\sqrt{T}}$. Hence, substituting the explicit expressions for $\eta_t = \frac{1}{\sqrt{dT}}$ and $\delta_t = \frac{1}{\sqrt{t}}$ into (22), dividing both sides with $T$, and using the convexity of function $F$, the desired bound can be concluded.

When $f_i \in \mathcal{F}_{\mathsf{lip}}(L_f, \mathcal{X}^\circ) \cap \mathcal{F}_{\mathsf{smo}}(s_f, \mathcal{X}^\circ)$, it can be shown that the term $(2NL_f \sum_{t=1}^{T} \delta_t)$ on the right-hand side of (22) is replaced by $(Ns_f \sum_{t=1}^{T} \delta_t^2)$, because of (20). As we discussed in the case of $f_i \in \mathcal{F}_{\mathsf{lip}}(L_f, \mathcal{X}^\circ)$, the convergence rate is determined by the terms involving $\frac{1}{\eta_t}$ and $\eta_t T$. Hence, the convergence rate is the same as that of the case when $f_i$ in only Lipschitz continuous. The proof is complete. $\square$

## C  PROOFS OF THEOREM 3 AND 4

First, we claim that for the DistZOOs $\tilde{\mathbf{g}}_i^{\mathsf{OP}}(\cdot)$ and $\tilde{\mathbf{g}}_i^{\mathsf{TP}}(\cdot)$, the strongly convexity of $f_i$ implies the strongly convexity of its smoothed variant $\hat{f}_i$, and its proof is straightforward. We now establish the basic convergence results for Algorithm 1 running with $\tilde{\mathbf{g}}_i^{\mathsf{OP}}(\cdot)$ and $\tilde{\mathbf{g}}_i^{\mathsf{TP}}(\cdot)$. It follows from (13) that

$$\mathbb{E}\left[\Lambda(t+1)\right] = \mathbb{E}\left[\Lambda(t)\right] + \eta_t^2 \sum_{i=1}^{N} \mathbb{E}\left[\|\tilde{\mathbf{g}}_i(\mathbf{x}_i(t); \delta_t)\|^2\right] - 2\eta_t \sum_{i=1}^{N} \mathbb{E}\left[\langle \nabla \hat{f}_i(\mathbf{x}_i(t); \delta_t), \mathbf{x}_i(t) - \mathbf{x}\rangle\right]$$
$$\leq \mathbb{E}\left[\Lambda(t)\right] + \eta_t^2 \sum_{i=1}^{N} \mathbb{E}\left[\|\tilde{\mathbf{g}}_i(\mathbf{x}_i(t); \delta_t)\|^2\right] - 2\eta_t \sum_{i=1}^{N} \left(\mathbb{E}\left[\hat{f}_i(\mathbf{x}_i(t); \delta_t)\right] - \hat{f}_i(\mathbf{x})\right)$$
$$- \mu_f \eta_t \mathbb{E}\left[\Lambda(t)\right] \tag{23}$$

where in the equality we used the relation $\mathbb{E}\left[\tilde{\mathbf{g}}_i(\mathbf{x}_i(t); \delta_t)\right] = \nabla \hat{f}_i(\mathbf{x}_i(t); \delta_t)$ (cf. Definition 1(i)) and in the inequality we used the strongly convexity of function $\hat{f}_i$. Summing the inequalities in (23) over $t = 1$ to $t = T$ and regrouping the terms, we obtain

$$\sum_{t=1}^{T} \sum_{i=1}^{N} \left(\mathbb{E}\left[\hat{f}_i(\mathbf{x}_i(t); \delta_t)\right] - \hat{f}_i(\mathbf{x})\right) \leq \frac{1}{2} \sum_{t=1}^{T} \eta_t \sum_{i=1}^{N} \mathbb{E}\left[\|\tilde{\mathbf{g}}_i(\mathbf{x}_i(t); \delta_t)\|^2\right]$$
$$+ \frac{1}{2} \sum_{t=1}^{T} \left(\frac{1}{\eta_t}\left(\mathbb{E}\left[\Lambda(t)\right] - \mathbb{E}\left[\Lambda(t+1)\right]\right) - \mu_f \mathbb{E}\left[\Lambda(t)\right]\right)$$
$$= \frac{1}{2} \sum_{t=1}^{T} \eta_t \sum_{i=1}^{N} \mathbb{E}\left[\|\tilde{\mathbf{g}}_i(\mathbf{x}_i(t); \delta_t)\|^2\right] + \frac{1}{2}\left(\frac{1}{\eta_1} - \mu_f\right) \mathbb{E}\left[\Lambda(1)\right]$$
$$+ \frac{1}{2} \sum_{t=2}^{T} \left(\frac{1}{\eta_t} - \frac{1}{\eta_{t-1}} - \mu_f\right) \mathbb{E}\left[\Lambda(t)\right] - \frac{1}{2\eta_T} \mathbb{E}\left[\Lambda(T+1)\right]. \tag{24}$$

By substituting the expression for $\eta_t = \frac{1}{\mu_f t}$ into (24) and dropping the negative term, it follows

$$\sum_{t=1}^{T} \sum_{i=1}^{N} \left(\mathbb{E}\left[\hat{f}_i(\mathbf{x}_i(t); \delta_t)\right] - \hat{f}_i(\mathbf{x})\right) \leq \frac{1}{2} \sum_{t=1}^{T} \eta_t \sum_{i=1}^{N} \mathbb{E}\left[\|\tilde{\mathbf{g}}_i(\mathbf{x}_i(t); \delta_t)\|^2\right]. \tag{25}$$

Then, following the same lines as that of the proof of Lemma 1, we have that, for DistZOOs $\tilde{\mathbf{g}}_i^{\mathsf{OP}}(\cdot)$ and $\tilde{\mathbf{g}}_i^{\mathsf{TP}}(\cdot)$,

$$\sum_{t=1}^{T} \left( \mathbb{E}\left[f(\mathbf{x}_{i\bullet}(t))\right] - f(\mathbf{x}^\star) \right) \leq \sum_{t=1}^{T} \mathbb{E}\left[\left|f(\mathbf{x}^\star) - \hat{f}(\mathbf{x}^\star; \delta_t)\right|\right] + \sum_{t=1}^{T} \mathbb{E}\left[\left|f(\mathbf{x}_{i\bullet}(t)) - \hat{f}(\mathbf{x}_{i\bullet}(t); \delta_t)\right|\right]$$

$$+ p_1 L_f + \frac{1}{2} \sum_{t=1}^{T} \eta_t \sum_{i=1}^{N} \mathbb{E}\left[\|\tilde{\mathbf{g}}_i(\mathbf{x}_i(t); \delta_t)\|^2\right] + p_2 L_f \sum_{t=1}^{T-1} \eta_t \sum_{i=1}^{N} \mathbb{E}\left[\|\tilde{\mathbf{g}}_i(\mathbf{x}_i(t); \delta_t)\|\right]. \tag{26}$$

(i) We now derive the convergence rate results for Algorithm 1 running with DistZOOs $\tilde{\mathbf{g}}_i^{\mathsf{OP}}(\cdot)$. When $f_i \in \mathcal{F}_{\mathsf{lip}}(L_f, \mathcal{X}^\circ) \cap \mathcal{F}_{\mathsf{sc}}(\mu_f, \mathcal{X}^\circ)$, we combine the results in (17), (26) and Lemma 4(i) to get

$$\sum_{t=1}^{T} \left( \mathbb{E}\left[f(\mathbf{x}_{i\bullet}(t))\right] - f(\mathbf{x}^\star) \right) \leq p_1 L_f + 2NL_f \sum_{t=1}^{T} \delta_t + \frac{1}{2}NC^2 d^2 \sum_{t=1}^{T} \frac{\eta_t}{\delta_t^2} + p_2 NL_f Cd \sum_{t=1}^{T} \frac{\eta_t}{\delta_t}.$$

By substituting $\eta_t = \frac{1}{\mu_f t}$ and $\delta_t = \frac{1}{t^{1/3}}$ into the preceding inequality and using the convexity of $F$, it yields the following optimal bound

$$\frac{1}{T} \sum_{t=1}^{T} \left( \mathbb{E}\left[f(\mathbf{x}_{i\bullet}(t))\right] - f(\mathbf{x}^\star) \right) = \mathcal{O}\left(\frac{d^2}{T^{1/3}}\right).$$

When $f_i \in \mathcal{F}_{\mathsf{lip}}(L_f, \mathcal{X}^\circ) \cap \mathcal{F}_{\mathsf{smo}}(s_f, \mathcal{X}^\circ) \cap \mathcal{F}_{\mathsf{sc}}(\mu_f, \mathcal{X}^\circ)$, it follows from an argument similar to that of (21) that

$$\sum_{t=1}^{T} \left( \mathbb{E}\left[f(\mathbf{x}_{i\bullet}(t))\right] - f(\mathbf{x}^\star) \right) \leq p_1 L_f + Ns_f \sum_{t=1}^{T} \delta_t^2 + \frac{1}{2}NC^2 d^2 \sum_{t=1}^{T} \frac{\eta_t}{\delta_t^2} + p_2 NL_f Cd \sum_{t=1}^{T} \frac{\eta_t}{\delta_t}.$$

It can be proven that the choice of $\delta_t = \frac{1}{t^{1/4}}$ yields the optimal convergence rate, that is, $\mathcal{O}\left(\frac{d^2}{\sqrt{T}}\right)$.

(ii) The proof for Algorithm 1 running with DistZOO $\tilde{\mathbf{g}}_i^{\mathsf{TP}}(\cdot)$ can be obtained in a similar way by exploiting the properties of the DistZOO $\tilde{\mathbf{g}}_i^{\mathsf{TP}}(\cdot)$.

□

## D PROOF OF THEOREM 5

We provide the basic convergence result for each stage $k$, and we start by deriving a similar bound as that of Lemma 1 as follows:

$$\sum_{t=1}^{T^{(k)}} \left( \mathbb{E}\left[f(\mathbf{x}_{i\bullet}^{(k)}(t))\right] - f(\mathbf{x}^\star) \right) \leq \sum_{t=1}^{T^{(k)}} \mathbb{E}\left[\left|f(\mathbf{x}^\star) - \hat{f}(\mathbf{x}^\star; \delta^{(k)})\right|\right] + \sum_{t=1}^{T^{(k)}} \mathbb{E}\left[\left|f(\mathbf{x}_{i\bullet}^{(k)}(t)) - \hat{f}(\mathbf{x}_{i\bullet}^{(k)}(t); \delta^{(k)})\right|\right]$$

$$+ \sum_{t=1}^{T^{(k)}} \frac{\mathbb{E}\left[\Lambda^{(k),\star}(t)\right] - \mathbb{E}\left[\Lambda^{(k),\star}(t+1)\right]}{2\eta^{(k)}} + p_1^{(k)} L_f + \frac{1}{2} \sum_{t=1}^{T^{(k)}} \eta^{(k)} \sum_{i=1}^{N} \mathbb{E}\left[\|\tilde{\mathbf{g}}_i(\mathbf{x}_i^{(k)}(t); \delta^{(k)})\|^2\right]$$

$$+ p_2 L_f \sum_{t=1}^{T^{(k)}} \eta^{(k)} \sum_{i=1}^{N} \mathbb{E}\left[\|\tilde{\mathbf{g}}_i(\mathbf{x}_i^{(k)}(t); \delta^{(k)})\|\right] \tag{27}$$

where $\Lambda^{(k),\star}(t) = \sum_{i=1}^{N} \|\mathbf{x}_i^{(k)}(t) - \mathbf{x}^\star\|^2$ and $p_1^{(k)}$ satisfies the following bound, according to compactness of the set $\mathcal{X}$,

$$p_1^{(k)} = 2N \max_{i \in \mathcal{V}} \{\|\mathbf{x}_{\mathsf{avg}}^{(k)}(1) - \mathbf{x}_i^{(k)}(1)\|\} + \frac{2N\alpha\beta}{1-\beta} \left( \sum_{i=1}^{N} \|\mathbf{x}_i^{(k)}(1)\| \right) \leq \left( 4N + \frac{2\alpha\beta}{1-\beta}N^2 \right) R_{\mathcal{X}}. \tag{28}$$

The left-hand side of (27) can be further bounded by using the strongly convexity of $F$, that is,

$$\frac{1}{T^{(k)}} \sum_{t=1}^{T^{(k)}} \left( f(\mathbf{x}_{i^\bullet}^{(k)}(t)) - f(\mathbf{x}^\star) \right) \geq \left\langle \nabla f(\mathbf{x}^\star), \hat{\mathbf{x}}_{i^\bullet}^{(k)}(T^{(k)}) - \mathbf{x}^\star \right\rangle + \frac{N\mu_f}{2} \|\hat{\mathbf{x}}_{i^\bullet}^{(k)}(T^{(k)}) - \mathbf{x}^\star\|^2$$

Applying the first-order optimality condition to the preceding inequality, i.e., $\left\langle \nabla F(\mathbf{x}^\star), \mathbf{x} - \mathbf{x}^\star \right\rangle \geq 0$ for any $\mathbf{x} \in \mathcal{X}$, yields

$$\frac{1}{T^{(k)}} \sum_{t=1}^{T^{(k)}} \left( f(\mathbf{x}_{i^\bullet}^{(k)}(t)) - f(\mathbf{x}^\star) \right) \geq \frac{N\mu_f}{2} \|\mathbf{x}_{i^\bullet}^{(k+1)}(1) - \mathbf{x}^\star\|^2 \tag{29}$$

where the second inequality follows from the non-expansiveness of the Euclidean projection $\mathbf{proj}_{\mathcal{X}}(\cdot)$, and the last equality from Step 3 in Algorithm 2. Combining the inequalities (27), (28) and (29), we have for any $i^\bullet \in \mathcal{V}$,

$$\frac{N\mu_f}{2} \mathbb{E}[\|\mathbf{x}_{i^\bullet}^{(k+1)}(1) - \mathbf{x}^\star\|^2] \leq \frac{\sum_{i=1}^{N} \mathbb{E}[\|\mathbf{x}_i^{(k)}(1) - \mathbf{x}^\star\|^2]}{2\eta^{(k)} T^{(k)}} + \left( 4N + \frac{2\alpha\beta}{1-\beta} N^2 \right) L_f R_{\mathcal{X}} \frac{1}{T^{(k)}}$$

$$+ \frac{1}{T^{(k)}} \sum_{t=1}^{T^{(k)}} \mathbb{E}[|f(\mathbf{x}^\star) - \hat{f}(\mathbf{x}^\star; \delta^{(k)})|] + \frac{1}{T^{(k)}} \sum_{t=1}^{T} \mathbb{E}[|f(\mathbf{x}_{i^\bullet}^{(k)}(t)) - \hat{f}(\mathbf{x}_{i^\bullet}^{(k)}(t); \delta^{(k)})|]$$

$$+ \frac{1}{2T^{(k)}} \sum_{t=1}^{T^{(k)}} \eta^{(k)} \sum_{i=1}^{N} \mathbb{E}[\|\tilde{\mathbf{g}}_i(\mathbf{x}_i^{(k)}(t); \delta^{(k)})\|^2] + p_2 L_f \frac{1}{T^{(k)}} \sum_{t=1}^{T^{(k)}} \eta^{(k)} \sum_{i=1}^{N} \mathbb{E}[\|\tilde{\mathbf{g}}_i(\mathbf{x}_i^{(k)}(t); \delta^{(k)})\|].$$

$$\tag{30}$$

From (30) we find that the convergence depends on the properties of the DistZOOs, and we first derive the dimension-dependence error bounds for DistZOO $\tilde{\mathbf{g}}_i^{\mathsf{OP}}(\cdot)$ and the bounds for $\tilde{\mathbf{g}}_i^{\mathsf{TP}}(\cdot)$ naturally follows from the derivations.

For DistZOO $\tilde{\mathbf{g}}_i^{\mathsf{OP}}(\cdot)$, when $f_i \in \mathcal{F}_{\mathsf{lip}}(L_f, \mathbb{B}_{R_{\mathcal{X}}}) \cap \mathcal{F}_{\mathsf{sc}}(\mu_f, \mathbb{B}_{R_{\mathcal{X}}})$, it follows from (17), (30) and Lemma 4(i) that

$$\mathbb{E}\left[ \max_{i \in \mathcal{V}} \{\|\mathbf{x}_i^{(k+1)}(1) - \mathbf{x}^\star\|^2\} \right] \leq \mathbb{E}\left[ \max_{i \in \mathcal{V}} \{\|\mathbf{x}_i^{(k)}(1) - \mathbf{x}^\star\|^2\} \right] \frac{1}{\mu_f \eta^{(k)} T^{(k)}}$$

$$+ 4\frac{L_f}{\mu_f} \delta^{(k)} + 4\left( 2 + \frac{\alpha\beta}{1-\beta} N \right) \frac{L_f}{\mu_f} R_{\mathcal{X}} \frac{1}{T^{(k)}} + 2p_2 \frac{L_f}{\mu_f} Cd \frac{\eta^{(k)}}{\delta^{(k)}} + \frac{1}{\mu_f} C^2 d^2 \frac{\eta^{(k)}}{(\delta^{(k)})^2}. \tag{31}$$

On the other hand, we have

$$T^{(k)} = a^{k-1} T^{(1)}, \qquad \eta^{(k)} = \frac{1}{a^{k-1}} \eta^{(1)}, \qquad \delta^{(k)} = \frac{1}{b^{k-1}} \delta^{(1)} \tag{32}$$

This, together with inequality (31), gives

$$\mathbb{E}\left[ \max_{i \in \mathcal{V}} \{\|\mathbf{x}_i^{(k+1)}(1) - \mathbf{x}^\star\|^2\} \right] \leq \mathbb{E}\left[ \max_{i \in \mathcal{V}} \{\|\mathbf{x}_i^{(k)}(1) - \mathbf{x}^\star\|^2\} \right] \frac{1}{\mu_f \eta^{(1)} T^{(1)}}$$

$$+ \frac{1}{\left( \min\{a, b, \frac{a}{b}, \frac{a}{b^2}\} \right)^{k-1}} \times \left( 4\frac{L_f}{\mu_f} \delta^{(1)} + 4\left( 2 + \frac{\alpha\beta}{1-\beta} N \right) \frac{L_f}{\mu_f} R_{\mathcal{X}} \frac{1}{T^{(1)}} \right. \tag{33}$$

$$\left. + 2p_2 \frac{L_f}{\mu_f} Cd \frac{\eta^{(1)}}{\delta^{(1)}} + \frac{1}{\mu_f} C^2 d^2 \frac{\eta^{(1)}}{(\delta^{(1)})^2} \right).$$

By substituting $T^{(1)} = m = 1$, $\eta^{(1)} = \frac{4\min\{b, \frac{a}{b^2}\}}{3\mu_f}$ and $\delta^{(1)} = 1$ into the preceding relation, we arrive at

$$\mathbb{E}\left[ \max_{i \in \mathcal{V}} \{\|\mathbf{x}_i^{(k+1)}(1) - \mathbf{x}^\star\|^2\} \right] \leq \mathbb{E}\left[ \max_{i \in \mathcal{V}} \{\|\mathbf{x}_i^{(k)}(1) - \mathbf{x}^\star\|^2\} \right] \frac{3}{4\min\{b, \frac{a}{b^2}\}} + \frac{R_1}{\left( \min\{b, \frac{a}{b^2}\} \right)^{k-1}} \tag{34}$$

where we have used the fact that $\min\left\{a, b, \frac{a}{b}, \frac{a}{b^2}\right\} = \min\left\{b, \frac{a}{b^2}\right\}$, due to $a > b^2$ (because of $\frac{a}{b^2} > 1$) and $b > 1$, and

$$R_1 = 4\frac{L_f}{\mu_f}\delta^{(1)} + 4\left(2 + \frac{\alpha\beta}{1-\beta}N\right)\frac{L_f}{\mu_f}R_\mathcal{X}\frac{1}{T^{(1)}} + 2p_2\frac{L_f}{\mu_f}Cd\frac{\eta^{(1)}}{\delta^{(1)}} + \frac{1}{\mu_f}C^2d^2\frac{\eta^{(1)}}{(\delta^{(1)})^2}.$$

We next show by induction that

$$\mathbb{E}\left[\max_{i\in\mathcal{V}}\left\{\|\mathbf{x}_i^{(k)}(1) - \mathbf{x}^\star\|^2\right\}\right] \leq \frac{4\max\{R_1, R_\mathcal{X}^2\}}{h^{k-2}} \tag{35}$$

where $h = \min\left\{b, \frac{a}{b^2}\right\}$. For $k = 1$, we can use the following bound to deduce that inequality (35) holds, $\max_{i\in\mathcal{V}}\left\{\|\mathbf{x}_i^{(1)}(1) - \mathbf{x}^\star\|^2\right\} \leq 4R_\mathcal{X}^2 \leq 4h\max\{R_1, R_\mathcal{X}^2\}$. We then assume that inequality (35) holds for $k$ and show it holds for $k + 1$ as well,

$$\mathbb{E}\left[\max_{i\in\mathcal{V}}\left\{\|\mathbf{x}_i^{(k+1)}(1) - \mathbf{x}^\star\|^2\right\}\right] \leq \frac{3}{4h}\mathbb{E}\left[\max_{i\in\mathcal{V}}\left\{\|\mathbf{x}_i^{(k)}(1) - \mathbf{x}^\star\|^2\right\}\right] + \frac{R_1}{h^{k-1}}$$

$$\leq \frac{3\max\{R_1, R_\mathcal{X}^2\}}{h^{k-1}} + \frac{\max\{R_1, R_\mathcal{X}^2\}}{h^{k-1}} \leq \frac{4\max\{R_1, R_\mathcal{X}^2\}}{h^{k-1}}$$

which leads to the conclusion in (35). It is easy to verify that the total number of stages in Algorithm 2 is $k^\natural = \left\lfloor \log_a\left(\frac{T}{m} + 1\right)\right\rfloor$, and the final estimates returned by Algorithm 2 are $\bar{x}_i(T), i \in \mathcal{V}$. Hence, applying $k^\natural + 1$ to (35), we have

$$\mathbb{E}\left[\max_{i\in\mathcal{V}}\left\{\|\bar{x}_i(T) - \mathbf{x}^\star\|^2\right\}\right] \leq \frac{4\max\{R_1, R_\mathcal{X}^2\}}{h^{k^\natural+1-2}} \leq \frac{4h^2\max\{R_1, R_\mathcal{X}^2\}}{h^{\log_a\left(\frac{T}{m}+1\right)}} = \frac{4h^2\max\{R_1, R_\mathcal{X}^2\}}{\left(\frac{T}{m}+1\right)^{\frac{1}{\log_h(a)}}} \tag{36}$$

where we used the inequality that $k^\natural \geq \log_a\left(\frac{T}{m} + 1\right) - 1$. We are left to find the minimum of $\log_h(a) = \log_{\min\{b, \frac{a}{b^2}\}}(a)$, which is achieved when $b = \frac{a}{b^2}$. This yields the following final convergence rate, that is, $\mathbb{E}\left[\max_{i\in\mathcal{V}}\left\{\|\bar{x}_i(T) - \mathbf{x}^\star\|^2\right\}\right] \leq \frac{4b^2\max\{R_1, R_\mathcal{X}^2\}}{(T+1)^{1/3}} = \mathcal{O}\left(\frac{d^2}{(T+1)^{1/3}}\right)$, where the equality follows from $R_1 = \mathcal{O}\left(d^2\right)$.

When $f_i \in \mathcal{F}_{\mathsf{lip}}(L_f, \mathbb{B}_{R_\mathcal{X}}) \cap \mathcal{F}_{\mathsf{smo}}(s_f, \mathbb{B}_{R_\mathcal{X}}) \cap \mathcal{F}_{\mathsf{sc}}(\mu_f, \mathbb{B}_{R_\mathcal{X}})$, the dimension-dependence bound can be obtained in a similar fashion.

For DistZOO $\tilde{\mathbf{g}}_i^{\mathsf{TP}}(\cdot)$, when $f_i \in \mathcal{F}_{\mathsf{lip}}(L_f, \mathbb{B}_{R_\mathcal{X}}) \cap \mathcal{F}_{\mathsf{sc}}(\mu_f, \mathbb{B}_{R_\mathcal{X}})$, it follows from (17), (30) and Lemma 4(ii) that

$$\mathbb{E}\left[\max_{i\in\mathcal{V}}\left\{\|\mathbf{x}_i^{(k+1)}(1) - \mathbf{x}^\star\|^2\right\}\right] \leq \mathbb{E}\left[\max_{i\in\mathcal{V}}\left\{\|\mathbf{x}_i^{(k)}(1) - \mathbf{x}^\star\|^2\right\}\right]\frac{1}{\mu_f\eta^{(k)}T^{(k)}}$$

$$+ 4\frac{L_f}{\mu_f}\delta^{(k)} + 4\left(2 + \frac{\alpha\beta}{1-\beta}N\right)\frac{L_f}{\mu_f}R_\mathcal{X}\frac{1}{T^{(k)}} + \frac{L_f^2}{\mu_f}\left(cd + 2p_2\sqrt{c}\sqrt{d}\right)\eta^{(k)}. \tag{37}$$

By setting $b = a$ and then substituting $T^{(1)} = 1$, $\eta^{(1)} = \frac{4a}{3\mu_f}$ and $\delta^{(1)} = 1$ into the preceding inequality it follows that

$$\mathbb{E}\left[\max_{i\in\mathcal{V}}\left\{\|\mathbf{x}_i^{(k+1)}(1) - \mathbf{x}^\star\|^2\right\}\right] \leq \mathbb{E}\left[\max_{i\in\mathcal{V}}\left\{\|\mathbf{x}_i^{(k)}(1) - \mathbf{x}^\star\|^2\right\}\right]\frac{3}{4a} + \frac{R_2}{a^{k-1}} \tag{38}$$

where $R_2 = 4\frac{L_f}{\mu_f}\delta^{(1)} + 4\left(2 + \frac{\alpha\beta}{1-\beta}N\right)\frac{L_f}{\mu_f}R_\mathcal{X}\frac{1}{T^{(1)}} + \frac{L_f^2}{\mu_f}\left(cd + 2p_2\sqrt{c}\sqrt{d}\right)\eta^{(1)}$. Then, following an argument similar to that of part (i), we obtain

$$\mathbb{E}\left[\max_{i\in\mathcal{V}}\left\{\|\bar{x}_i(T) - \mathbf{x}^\star\|^2\right\}\right] \leq \frac{4a^2\max\{R_2, R_\mathcal{X}^2\}}{T+1} = \mathcal{O}\left(\frac{d}{T+1}\right).$$

Similarly, when $f_i \in \mathcal{F}_{\mathsf{lip}}(L_f, \mathbb{B}_{R_\mathcal{X}}) \cap \mathcal{F}_{\mathsf{smo}}(s_f, \mathbb{B}_{R_\mathcal{X}}) \cap \mathcal{F}_{\mathsf{sc}}(\mu_f, \mathbb{B}_{R_\mathcal{X}})$, we have

$$\mathbb{E}\left[\max_{i\in\mathcal{V}}\left\{\|\bar{x}_i(T) - \mathbf{x}^\star\|^2\right\}\right] \leq \frac{4a^2\max\{R_2', R_\mathcal{X}^2\}}{T+1} = \mathcal{O}\left(\frac{d}{T+1}\right)$$

where $R_2' = 2\frac{s_f}{\mu_f}(\delta^{(1)})^2 + 4\left(2 + \frac{\alpha\beta}{1-\beta}N\right)\frac{L_f}{\mu_f}R_\mathcal{X}\frac{1}{T^{(1)}} + \frac{L_f^2}{\mu_f}\left(cd + 2p_2\sqrt{c}\sqrt{d}\right)\eta^{(1)}$. The proof is complete. $\qquad\square$

