# OpenReview forum: "Distributed Zeroth-Order Optimization: Convergence Rates That Match Centralized Counterpart"
_ICLR.cc/2022/Conference — ICLR 2022 Submitted_

### Official Review · Reviewer_fRi4 · 2021-11-03

**Correctness:** 3
**Technical Novelty And Significance:** 2
**Empirical Novelty And Significance:** 2
**Recommendation:** 5
**Confidence:** 4

**Main Review:**

0. This paper is well organized and clearly written. The results seem to be reasonable.
1. The proposed algorithms look like simple extensions from the first-order to the zeroth-order. The authors should highlight the novel contributions of this paper. Are there any particular challenges in analyzing the zeroth-order algorithms, compared to analyzing the first-order ones?
2. The communication graph should play an important role in the established convergence rates, but the main theorems fail to clarify this issue. Detailed discussions would be helpful.
3. In the first paragraph, the authors motivate with applications like adversarial attacks in deep learning and policy search in reinforcement learning. However, the numerical experiments end up with a simple least squares problem. More extensive numerical experiments are necessary.


**Summary Of The Paper:**

In this paper, the authors develop a distributed zeroth-order algorithm over a time-varying communication graph, as well as its multi-stage variant with time-varying step size. Convergence rates are established and compared with those of the centralized algorithms.

**Summary Of The Review:**

Overall, this paper has disadvantages in novelty, depth of analysis, and numerical experiments.

---

> ### Author Response · Authors · 2021-11-19
> **The dependence on network connectivity has been established and more comparisons with first-order algorithms will be made**
>
> We thank the reviewer for the time in evaluating our work and the in-depth comments.
>
> We will further highlight the contributions and technical difficulties of the distributed zeroth-order algorithms, as compared to the first-order algorithms. In fact, this paper aims to fill the gap between centralized and distributed zeroth-order optimization and provide a unified analysis of the distributed zeroth-order optimization that is based on gradient-free oracles. Any zeroth-order oracles that fit into Definition 1 will naturally follow from our analysis.
>
> As for the communication, we have established the dependence of the error bounds on the time-varying network structure, which has been encoded in the parameter $\beta$ (see Lemma 1). We shall also highlight the dependence on $\\beta$ in Table 1.
>
> More discussions about the effects of the network topology on convergence will be added. Finally, numerical experimental will be revised by considering the real-world examples as well.

---

### Official Review · Reviewer_5m6d · 2021-11-08

**Correctness:** 3
**Technical Novelty And Significance:** 3
**Empirical Novelty And Significance:** 2
**Recommendation:** 3
**Confidence:** 4

**Main Review:**

Strength:
It seems that this paper provided new convergence results for the decentralized setting.

Weakness:
1. However, the related works are not well-compared or missing. So it's hard for me to judge the results of this paper. For instance, in Table 1,
- the results for the Lipschitz class clearly also hold for Lipschitz and smooth class;
- the non-accelerated result for the smooth class is $O(\frac{d}{T})$ (see, e.g., [1, 2] via directional derivative), and also accelerated result $O(\frac{d^2}{T^2})$ is obtained in [2] via two-point gradient estimators. Both results are much better than the one in the current submission, but the authors did not list and compare with them.
- for the strongly convex class, the convergence rate is linear (e.g., [2] provides the accelerated rates $\Big(1-\frac{\sqrt{\mu_f/s_f}}{d}\Big)^T$ ), while the authors listed the much weaker sublinear results in Table 1.
-ps: I do not think the constrained/unconstrained domain will affect the comparison with results of [2], since the current submission directly uses the non-expansiveness of the Euclidean projection **proj**($\cdot$) in their proofs (see Page 11-12)).


2. The authors consider the decentralized setting, however, their convergence results (see Table 1 and Theorems 1-4)  do not depend on any parameters (e.g., spectral gap) regarding the decentralized network topology. so it's also hard for me to believe the correctness of their results.



[1] Arkadi Nemirovski, Anatoli Juditsky, Guanghui Lan, and Alexander Shapiro. Robust stochastic approximation approach to stochastic programming. *SIAM Journal on optimization*, 19(4):1574–1609, 2009.

[2] Yurii Nesterov and Vladimir Spokoiny. Random gradient-free minimization of convex functions. *Foundations of Computational Mathematics*, 17(2):527–566, 2017.

**Summary Of The Paper:**

This paper studied zeroth-order methods via one-point and two-point gradient estimators for decentralized optimization.

**Summary Of The Review:**

The comparison of results is not appropriate and accurate. Also, their results look strange (e.g., do not depend on the parameter of network topology). So I recommend a reject.

---

> ### Author Response · Authors · 2021-11-19
> **Table 1 will be updated and more clarifications will be added**
>
> We thank the reviewer for the time in evaluating our work and the constructive comments.
>
> It is worth noting that the authors in [2] utilize the Gaussian smoothing parameters to construct the gradient estimator, this is fine for unconstrained domains. However, for constrained optimization, the decision variable ($x_i(t) + \delta_t u_i$) may not lie in the constraint set, and the corresponding assumption about the Lipschitz continuity over the domain does not apply. But the reviewer raised a very nice point, we will update the convergence results in Table 1 by doing a more thorough literature review. And we will also try to present refined convergence rates for the case of strongly and smooth optimization, however, we would like to emphasize that there may exist some gap between the centralized and distributed optimization.
>
> Indeed, we have established the dependence of the error bounds on the time-varying network structure, which has been encoded in the parameter $\beta$ (see Lemma 1). It is straightforward to highlight the dependence of the final convergence bounds in Table 1 on $\beta$. We will fix this.

---

### Official Review · Reviewer_JpXi · 2021-11-08

**Correctness:** 3
**Technical Novelty And Significance:** 3
**Empirical Novelty And Significance:** 2
**Recommendation:** 6
**Confidence:** 5

**Main Review:**

This paper is well written and the theoretical results are solid. I have a few concerns as follows,

- The novelty of this paper: the idea of both the MOZAPA and multi-stage MOZAPA algorithms are not novel. They all have their first-order counterparts and have been presented years ago. Although the authors do give credits to these previous work, the novelty of the two algorithms should be doubted as the only change in the algorithms is to replace the first order oracle with a zeroth-order one.

- Some assumptions may need some explanation. In theorem 1 and 3, the authors present their results based on the assumption that $|f_i(x_i(t) + \delta_t u_i(t))| \leq C$, for all $i \in \mathcal{V}$. I understand this assumption is needed for one-point oracle, but we do need some explanation and comparison between the assumptions for OP oracle and TP oracle.

- The comparison table on Page 3 presents some sub-optimal results for the centralized counterpart. For example, for a two-point oracle with strong convexity, the optimal rate should be $\mathcal{\tilde{O}}(d/T)$, if one refers to some previous work like Duchi et al. (2015). The sub-optimal rate should not be presented.

- The numerical experiments are trivial and need improvement. The experiment is a toy example, with these theoretical findings, we do not expect very comprehensive numerical results, but at least run 100 MC simulations, instead of 10. We also need a comparison with the centralized zeroth-order algorithm.

**Summary Of The Paper:**

This paper proposes a zeroth-order optimization algorithm for distributed, multi-agent systems with time-varying communication networks. The authors show that their presented multi-agent zeroth-order projection averaging algorithm (and its improved multi-stage version) has a convergence rate that matches the centralized counterpart algorithms under different assumptions. A small numerical experiment is also conducted to illustrate their theoretical findings.

**Summary Of The Review:**

This paper presents some rigorous theoretical findings for the distributed zeroth-order SGD. However, my main concern is the novelty of this paper. There are some modifications needed on the assumtion and numerical experiments as well.

---

> ### Author Response · Authors · 2021-11-19
> **Thank you for your constructive comments**
>
> We thank the reviewer for the time in evaluating our work and the constructive comments.
>
> Indeed the idea of both the MOZAPA and multi-stage MOZAPA algorithms have been in the literature as we acknowledged in the manuscript. They are arguably the most basic consensus-based distributed algorithms, and therefore a thorough convergence rate analysis, especially over time-varying networks, is of interest but missing. We aim to fill this gap by our work.
>
> We will update the convergence results in Table 1 by taking Duchi’s work into account. And numerical experimental results will be improved as well, by running at least 100 MC simulations and adding more comparisons with the centralized benchmark algorithms.

---

### Decision · Program_Chairs · 2022-01-20

**Decision:**

Reject

**Comment:**

The paper is about a topic that has been extensively studied for more than a decade, hence a very precise discussion of prior work as well as the new insights is absolutely necessary. Unfortunately both are lacking at this stage, thus the paper cannot be accepted.